# Quality of prescribing in older people from a broad family physician perspective: a descriptive pilot study

Naldy Parodi López,[1,2] Susanna Maria Wallerstedt[2,3]

¹Närhälsan Kungshöjd Health Centre, Gothenburg, Sweden
²Department of Pharmacology, Sahlgrenska Academy, University of Gothenburg, Gothenburg, Sweden
³HTA-centrum, Sahlgrenska universitetssjukhuset, Gothenburg, Sweden

**Correspondence to**
Dr Naldy Parodi López;
naldy.parodi.lopez@vgregion.se

## ABSTRACT

**Objectives** To investigate the quality of drug treatment in older people from a broad family physician perspective, and to provide evidence for power calculations in full-scale studies on prescribing quality.

**Design** Descriptive, retrospective pilot study.

**Setting** A primary healthcare centre in Sweden.

**Participants** 123 consecutive patients, ≥65 years, with a non-urgent physician consultation in January 2016.

**Measures** The drug treatment was assessed by a physician as either appropriate or suboptimal, taking individual factors like morbidity, life expectancy and concurrent drug treatment into account, and preceded by the application of 493 criteria from three screening tools for Potentially Inappropriate Medications (PIMs) and Potential Prescribing Omissions (PPOs). Suboptimal drug treatment was further categorised regarding priority: (1) immediate change suggested or (2) actions suggested in the longer term. Prevalence of the procedure code 'medication review' and the results thereof were also recorded.

**Results** Median age: 76 years; 48% women. When a family physician perspective was applied, and 593 PIMs/PPOs identified in 117 (95%) patients considered, 45 (37%) patients had suboptimal drug treatment. Immediate handling was suggested in 13 (11%) patients, most often concerning withdrawals of drugs for anxiety and insomnia. Handling in the longer term was suggested in 32 (26%) patients, most often concerning overuse of proton pump inhibitors. Over the last year, the procedure code 'medication review' was recorded for 65 (53%) patients. In medication reviews recorded during January 2016 (n=45), 23 (7%) drugs out of 309 were acted on, most often a dosage adjustment.

**Conclusions** This pilot study shows that when a broad family physician perspective is applied, taking individual factors and medical priorities in the complex clinical situation into account, drug treatment in primary care is appropriate for the majority of older patients. The results may be useful in sample size considerations for future studies on prescribing practices.

## BACKGROUND

The purpose of drug treatment is to increase the health of patients. However, prescribing of drugs is a challenge, particularly in older people who are sensitive to drug effects and often suffer from multiple morbidities.

### Strengths and limitations of this study

► This pilot study contributes current and clinically relevant information on the quality of drug treatment in older people from a broad family physician perspective, taking patient factors and medical priorities in the complex clinical situation into account, thereby providing figures for sample size consideration in full-scale studies on prescribing quality.

► To systemise the physician assessments and ensure that important potentially inappropriate medications and potential prescribing omissions were not overlooked, the assessments were preceded by the application of numerous screening tools.

► The limited sample size and the inclusion of one primary healthcare centre only, reduce the external validity and full-scale studies should include two assessors to allow estimations of inter-rater agreement.

Numerous studies have focused on describing and improving prescribing practices in this age group.[1–5]

In order to improve the quality of drug treatment, and thereby improving patient health, some countries have incorporated medication reviews in guidelines.[6–9] Frequently, medication reviews are performed by pharmacists.[3 10] In Sweden, the National Board of Health and Welfare decided in 2012 that medication reviews lie within the professional responsibilities of the physician; the only profession with a full licence to prescribe. According to national regulations, a medication review implies that the physician reconciles the drug treatment and assesses the benefit/risk for all drugs separately and combined, to ascertain that the treatment is reasonable given the current health status.[11] Medication reviews are to be documented in the medical records by a procedure code. In the Region Västra Götaland, where this study was performed, registration of this code in primary care results in monetary compensation.

Interestingly, numerous systematic reviews have failed to demonstrate effects of

specifically organised medication reviews on the outcomes mortality and hospitalisations.[12–16] Provided that inappropriate prescribing is a major problem in healthcare,[4 5 17] this may be surprising. If, on the other hand, prescribing quality problems are less pronounced than the scientific literature may suggest, the lack of patient relevant effects may be less surprising. In this context, it is important to note that studies investigating the quality of drug treatment often focus on the prevalence of Potentially Inappropriate Medications (PIMs) and Potential Prescribing Omissions (PPOs), and 3 in 10 PIMs as well as 7 in 10 PPOs have been shown not to be clinically relevant at the individual level.[18] Therefore, to learn more about the quality of current prescribing practices, a physician approach, based on an overall medical assessment of the information available for a specific patient, would add important information.

Prioritisations may be an additional aspect of importance when evaluating prescribing quality concerns. As older people often have a complex clinical situation, physicians in constraints of time may have to prioritise the most urgent needs of the patient. To the best of our knowledge, this perspective has not previously been described in the scientific literature. Further, as far as we are aware, drug treatment changes on, and documentation of, medication reviews by the attending physicians in daily care, have not previously been described.

The aim of this pilot study, with the underlying goal to provide evidence for power calculations in full-scale studies on prescribing quality, was to investigate the quality of drug treatment in older people from a broad family physician perspective, taking individual factors and medical priorities in the complex clinical situation into account. We also wanted to describe prescribing practices by physicians in primary care, with focus on medication reviews as performed according to a procedure code in the medical record.

## MATERIALS AND METHODS
### Design and setting
A descriptive pilot study was performed in a primary healthcare centre belonging to the Swedish National Health Service in Region Västra Götaland. The centre serves approximately 10 000 patients and is staffed by about 10 physicians (five specialists in family medicine, three residents in family medicine, two licensed physicians without specialist competence and one intern). Two nursing homes, with approximately 100 residents, are attached to the centre.

### Participants
All patients, ≥65 years of age, with a non-urgent consultation registered in the primary healthcare centre in 1 January to 31 January 2016, were included in the study. Thus, visits in the healthcare centre as well as visits in nursing homes and home visits were consecutively included if not caused by an urgent event. If a patient

had >1 consultation during the study period, the first one, as well as the information available prior to this visit, was included in the study.

### Data source
Data were retrospectively and manually extracted from electronic medical records (Asynja Visph and Journal III). The centre physicians were not informed beforehand that their work would be monitored the month in question. For each patient, socio-demographic and health-related data were retrieved including age, sex, residence, cognition, multi-dose drug dispensing (machine-dispensed unit bags with drugs that should be ingested concomitantly, for patients who have difficulties in handling their medications), drug treatment and morbidities. Regarding morbidities, we recorded the presence of medical conditions appearing in the Screening Tool of Older Persons' potentially inappropriate prescriptions (STOPP), the Screening Tool to Alert to Right Treatment (START),[19] the EU (7)-PIM list[20] and the set of indicators of prescribing quality provided by the Swedish National Board of Health and Welfare.[21]

For each drug in the current medication list, at the end of the visit according to the medical records including electronic prescriptions with detailed information on prescribed products and doses, we recorded the substance name, the Anatomic Therapeutic Classification (ATC) code,[22] and information as to whether the drug was prescribed for regular use or as needed. Drugs for external use were included only if having potential systemic effects. Combination drugs were counted as one drug. Two drugs including the same substance and formulation, for example, two different dosages of the same drug, were counted as one drug. The number of drugs that were used regularly (ie, not for a temporary condition such as an infection) was recorded.

We recorded if a procedure code, explicitly stating that a medication review had been performed, was present or absent in the medical records of the physician consultation in January 2016. If present, we recorded the actions taken and the content of the related documentation, based on the information available in the medical records and the electronic prescriptions. Thus, each drug in the medication list was categorised as unchanged, withdrawn, dosage adjusted, added or other. The changes in drug treatment were also summated at the patient level, categorised as ≥1 drugs being withdrawn, dosage adjusted, added, or other. Further, we recorded how the actions were documented in the medical records, focusing on whether the drug treatment problem was described, the goal with the treatment mentioned and planned follow-up expressed. In order to elucidate to what extent the procedure code medication review is documented at least annually, we extended the search in the medical record for this action, if not recorded in January 2016, to the entire year preceding the physician consultation. The contents of these medication reviews were not further

analysed, as the focus of the study was drug treatment as of January 2016.

## Assessments

### From a family physician perspective

For every patient, the quality of the drug treatment was assessed at the overall level, from the perspective of a physician specialised in family medicine (NPL), with experience of assessing drug treatment quality in older people,[23] taking into account individual characteristics of the patient such as morbidity, life expectancy and concurrent drug treatment. The assessments were based on the information in the electronic medical records and the assessor's clinical experience. The assessor categorised each patient's drug treatment as either *appropriate* or *suboptimal*. Suboptimal treatment was further categorised according to the suggested handling of the drug treatment change: *high priority* if the suggestion was to change ≥1 drugs immediately, and *low priority* if changes could be handled in the longer term. In cases of doubt concerning the assessments, a senior specialist physician in clinical pharmacology (SMW) was consulted, both authors had a discussion and came to a consensus decision. Information on suboptimal treatment prioritised for an immediate action was forwarded to the physician who attended the patient.

### Application of screening tools prior to physician assessment

In order to systemise the physician assessments and ensure that important PIMs and PPOs were not overlooked, the overall assessment of each patient was preceded by the manual application of a total of 493 criteria from three established screening tools: the STOPP/START criteria V.2, consisting of 80 PIMs and 34 PPOs,[19] the EU(7)-PIM list consisting of 282 PIMs[20] and the Swedish indicator set in which we used 97 out of 98 drug-specific and diagnosis-specific indicators of PIMs (n=77) as well as rational (n=20) treatments/treatment strategies[21]; one diagnosis-specific indicator was excluded because it was not applicable at the individual level. To comply with the other indicator sets, we recorded the absence of a rational treatment/treatment strategy in the Swedish indicator set as a PPO. The sets partly overlapped as, for example, potential overuse of proton pump inhibitors (PPI) and long-acting benzodiazepines were alerted in all sets. Renal function, reflected in the estimated glomerular filtration rate (eGFR), was extracted from the medical records. For patients where an eGFR was not provided in this source, we calculated creatinine clearance according to the Cockcroft-Gault equation. The assessing physician had a hard copy with all criteria, and all detected PIMs/PPOs were marked for each patient. The assessor was also allowed to identify inappropriate drug treatment not covered by the screening tools.

## Statistics

SPSS (IBM SPSS Statistics for Windows, V.19.0) was used for descriptive analyses concerning characteristics of patients and drug treatment by the absence/presence of ≥1 procedure codes for medication review over the last year. In the sample size considerations for this pilot study, we considered that including all patients fulfilling our inclusion criteria during 1 month, anticipated to amount to about a hundred, would yield useful results for the power calculations in the full-scale study, that is, reasonably certain prevalence figures on suboptimal drug treatment as well as information on the prevalence of documented medication reviews. Values are presented as numbers (percentages) as well as mean±SD or median and IQR or range where appropriate. As the Swedish regulations declare that physicians shall perform annual medication reviews for persons ≥75 years of age with ≥5 drugs in the medication list, these categories were also used.

## Patient and public involvement

This research was done without patient involvement. Patients were not invited to comment on the study design and were not consulted to develop patient relevant outcomes or interpret the results. Patients were not invited to contribute to the writing or editing of this document for readability or accuracy.

## RESULTS

A flowchart of the study population is presented in figure 1. A total of 123 patients were included in the analyses, 59 (48%) of whom were female (table 1). The median age was 76 years, ranging from 65 to 102 years. The number of chronic drugs was 5 (IQR: 3–7). Impaired cognition including dementia was found in 24 (20%) patients, and 24 (20%) individuals were nursing home residents. The most common medical conditions were hypertension (n=89; 72%), chronic pain (n=50; 41%), type 2 diabetes mellitus (n=39; 32%) and osteoarthritis (n=35; 28%).

In all, the medication lists of all patients contained 773 drugs. The most commonly prescribed drugs were paracetamol (44 patients; 34%), acetylsalicylic acid (38 patients; 30%), atorvastatin (29 patients; 24%), enalapril (29 patients; 24%), cyanocobalamin (28 patients; 23%) and metformin (27 patients; 22%).

According to the physician assessment, 78 (63%) patients had *appropriate* treatment and 45 (37%) patients had *suboptimal* drug treatment, that is, a change was suggested (table 2). In 13 patients (11% of all; 29% of those with suboptimal treatment), the suggested change in drug treatment was of high priority, ie, suggested to take place immediately, and concerned mainly withdrawal of propiomazine (an atypical antipsychotic agent with sedative properties used for insomnia) and hydroxyzine (a first-generation antihistamine primarily used for anxiety). In the 32 remaining cases (26% of all, 71% of those with suboptimal treatment), the suggested change was of lower priority, that is, the problem/s/was suggested

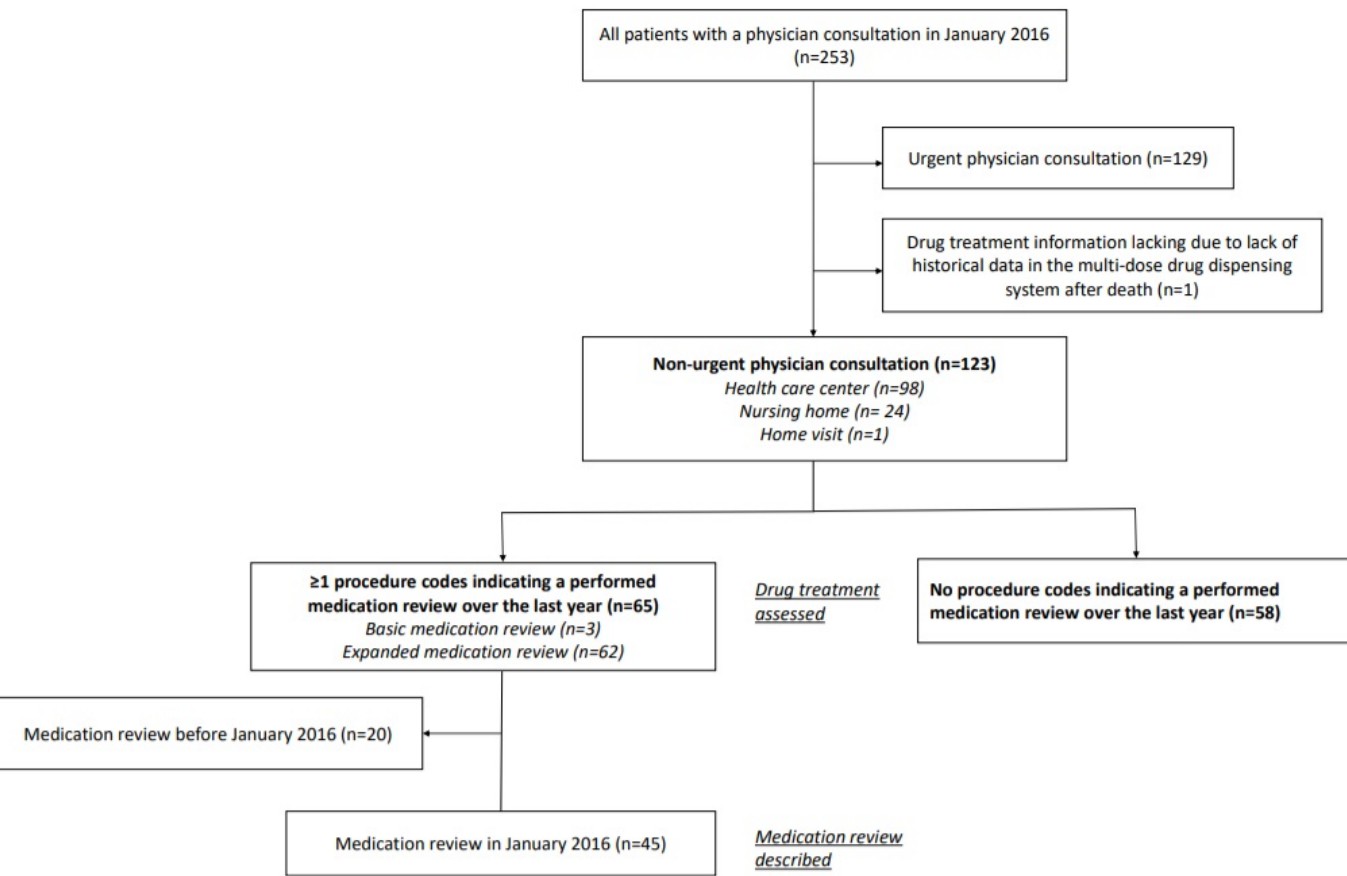

**Figure 1** Flow chart of the study population.

to be handled in the longer term, and concerned mainly potential overtreatment with PPI.

The application of screening tools prior to the physician assessment resulted in the identification of 593 PIMs/PPOs (median: 4; IQR: 2–7), partly overlapping, in 117 (95%) patients. STOPP identified 125 PIMs in 59 (48%) patients (median: 2; IQR: 1–3), and START identified 54 PPOs in 49 (40%) patients (median: 1; IQR: 1–1). The EU(7)-PIM list identified PIMs in 58 (47%) patients (median: 1; IQR: 1–2). The Swedish indicator set identified 96 PIMs in 48 (39%) patients (median: 2; IQR: 1–3) and 226 PPOs in 105 (85%) patients (median: 2; IQR: 1–3).

A total of 65 patients (53%) had ≥1 procedure codes stating that a medication review had been performed over the last year. Among those ≥75 years of age with ≥5 drugs in the medication list (n=53), 41 (77%) patients had ≥1 procedure codes registered regarding a medication review performed over the last year. The proportion of patients with suboptimal drug treatment according to the physician assessment was equally distributed between patients with and without a procedure code for a medication review performed over the last year: 24 (37%) and 21 (36%), respectively. Six of 13 patients with suboptimal treatment of high priority had a procedure code in the medical record explicitly stating that the drug treatment had been reviewed at least once over the last year. Among 32 patients with suboptimal treatment with low priority,

18 had ≥1 procedure codes regarding a medication review over the last year.

For 15 (33%) out of 45 patients with a procedure code for a medication review in January 2016, ≥1 actions were documented in the medical record, most often a dosage adjustment (table 3). These actions concerned 23 (7%) drugs out of 309 present in the medication lists of these patients, and a total of 14 active substances. In all, three actions were comprehensively documented in the medical records regarding the underlying problem, the treatment goal and the planned follow-up. There were no changes in drug treatment due to reduced renal function.

## DISCUSSION
### Main findings
In this study, we show that more than 6 in 10 patients in primary care had appropriate drug treatment, when assessed from an overall perspective by a physician, taking morbidity, life expectancy and concurrent drug treatment into account. Only 1 in 10 patients had suboptimal drug treatment where an immediate change was suggested to be prioritised, mostly concerning a withdrawal of sedatives/anxiolytics. For one in four patients, the suboptimal treatment was suggested to be of lower priority, predominantly concerning the withdrawal of a PPI.

According to the procedure codes in the medical records, primary care physicians documented medication

**Table 1** Characteristics of patients according to the presence/absence of ≥1 procedure codes indicating a performed medication review over the last year

| | ≥1 Procedure codes for a medication review over the last year | |
|---|---|---|
| | Yes (n=65) | No (n=58) |
| Age | | |
| Years | 79 (73–84) | 73 (68–79) |
| ≥75 years | 45 (69) | 19 (33) |
| Sex, female | 30 (46) | 29 (50) |
| Impaired cognition, including dementia | 16 (25) | 8 (14) |
| Nursing home resident | 15 (23) | 9 (16) |
| Multi-dose drug dispensing | 11 (17) | 6 (10) |
| Common morbidities in the sample | | |
| Hypertension | 54 (83) | 35 (60) |
| Type 2 diabetes mellitus | 34 (52) | 5 (9) |
| Chronic pain | 30 (46) | 20 (35) |
| Gastro-oesophageal reflux disease or peptic ulcer disease | 17 (26) | 10 (17) |
| Osteoarthritis | 16 (25) | 19 (33) |
| Insomnia | 13 (20) | 13 (22) |
| Cardiovascular disease | 12 (19) | 8 (14) |
| eGFR<60 mL/min | 12 (19) | 6 (10) |
| Dementia | 11 (17) | 5 (9) |
| Depression | 11 (17) | 11 (19) |
| Anxiety | 9 (14) | 10 (17) |
| Number of drugs post-visit | | |
| Total | 7 (6–10) | 5 (3–6) |
| Regular | 7 (5–8) | 4 (2–6) |
| As needed | 1 (0–2) | 1 (0–1) |
| ≥5 Drugs | 54 (83) | 30 (52) |

Values are presented as median (IQR), or number of individuals (percentage).
eGFR, estimated glomerular filtration rate.

reviews in about half of all older people at least annually, with focus on those of old age with extensive comorbidities and long medication lists. For one in three patients, the medication review resulted in an active change of treatment, most often a dosage adjustment as, for example, increasing the metformin dose due to high glycated haemoglobin.

### Strengths and limitations

To the best of our knowledge this is the first study to provide information on the quality of drug treatment in older people in primary care from a broad family physician perspective, taking individual factors and priorities in the complex clinical situation into account. Thereby, our pilot study contributes valuable information for sample size consideration in future studies on prescribing practices and interventions for improved drug treatment. Another strength is that the physician assessment was preceded by the application of numerous screening tools for PIMs and PPOs, thus minimising the risk of overlooking important aspects regarding the prescribing quality. Indeed, only 1 in 20 patients did not have any PIMs/PPOs, indicating that our screening tools captured potentially suboptimal treatment to a greater extent than many other studies.[4 5 17] An additional strength is the fact that only one patient was excluded, an important aspect for generalisability.

Limitations of this study include the sample size and the fact that data were obtained from one healthcare centre only. This may reduce the external validity as centres may differ from each other regarding, for instance, population covered and prescribing traditions.[24] However, a limited sample size may be justified in a pilot study; a full-scale study, sized on the basis of these pilot results, has been initiated. Another limitation is that the physician assessments were performed by one person and not a team. However, this physician had relevant expertise as well as previous experience of scientifically assessing pharmacotherapy in older people.[23] In addition, ambiguities in the assessments were discussed with a senior clinical pharmacologist. Nevertheless, full-scale studies should preferably include two assessors and consensus discussions to provide information on inter-rater agreement. Another limitation is that the study was restricted to information in the medical records from primary care. Indeed, the assessments were based on available data in routine care, carrying a risk that the information about the medical history, diagnoses, examinations and laboratory tests had not been documented or appropriately coded in the medical records. Yet, this data source is more comprehensive than drug registers, either alone or linked to other registers, which are frequently used in studies on prescribing practices.[25]

### Interpretation

The apparent discrepancy between the prevalence of suboptimal drug treatment according to the physician assessment and the screening tools illustrates the importance of taking future research on prescribing practices a step further to increase the medical relevance. Our results suggest that the limitations regarding the concurrent validity of general screening tools in hip fracture patients and the very old[18 26–28] may also apply to older people in general in primary care. Indeed, although the positive predictive value for several indicator sets has been reported to be acceptable when determined separately,[26 27] the PIMs/PPOs may not be relevant when taking the overall clinical picture into account, that is, from a family physician perspective.

The low prevalence of suboptimal drug treatment where an immediate change was suggested contributes to the understanding of the medical practice in primary care. In fact, the level of priority, according to

**Table 2**  Suggested drug changes in patients with suboptimal treatment according to the physician assessment

| Priority | Suggestion | Drug (n) |
|---|---|---|
| High, change immediate | Withdrawal | Propiomazine (n=5), hydroxyzine (n=4), diazepam (n=1), naproxen (n=1), nifedipine (n=1), orphenadrine (n=1), tramadol (n=1) |
| Low, reconsider in the longer term | Withdrawal, apparent indication lacking or probably inappropriate treatment | PPI (n=18), cyanocobalamin (n=2), metformin (n=2), acetylcysteine (n=1), acetylsalicylic acid (n=1), biperiden (n=1), diclofenac (n=1), dipyridamole (n=1), furosemide (n=1), laxative (n=1), potassium (n=1), quetiapine (n=1), risperidone (n=1), sotalol (n=1), zopiclone (n=1), felodipine/carbamazepine interaction (n=1) |
|  | Addition, absence of recommended treatment | Bisphosphonate (n=1), laxatives (n=1) |

PPI, proton pump inhibitors.

our definition, may represent a balance between feasibility and expected patient harm; changes that can be achieved with reasonable efforts in relation to the anticipated patient benefit will get a higher priority than the opposite. Indeed, the changes suggested by the assessing physician predominantly concerned sedatives/anxiolytics and PPIs, withdrawals of which may not easily be achieved once they have been initiated.[24 29] The high prevalence of PPI without an underlying indication is in agreement with a recent population-based register-based study where no underlying disease-related or drug-related reason could be identified for 4 in 10 patients on long-term treatment with PPI.[30] Our pilot findings also contributes to the understanding of the limited implementation of recommendations for drug changes after medication reviews by third parties.[31]

**Table 3**  Actions taken and contents of documentation in the medical records during documented medication reviews in January 2016 (n=45)

| | Patients* n (%) | Drugs involved | Documentation in the medical records |
|---|---|---|---|
| **Actions taken** | | | |
| Drug withdrawn | 3 (7) | Omeprazole (n=2) | P/G (n=1); F (n=1) |
| | | Felodipine (n=1) | P (n=1) |
| | | Sertraline (n=1) | P/G (n=1) |
| | | Zopiclone (n=1) | P/G (n=1) |
| Drug dosage adjusted | 10 (22) | Felodipine (n=1) | P (n=1) |
| | | Insulin (n=3) | P (n=1); P/G (n=2) |
| | | Metformin (n=2) | P (n=1); P/G/F (n=1) |
| | | Metoprolol (n=1) | P (n=1) |
| | | Propiomazine (n=1) | P/G (n=1) |
| | | Repaglinide (n=1) | P/G/F (n=1) |
| | | Sertraline (n=1) | P/G (n=1) |
| Drug added | 4 (9) | Omeprazole (n=1) | P/G (n=1) |
| | | Atorvastatin (n=1) | P/F (n=1) |
| | | Enalapril (n=1) | P/G/F (n=1) |
| | | Hydrochlorothiazide (n=1) | P (n=1) |
| | | Melatonin (n=1) | P/G (n=1) |
| | | Paracetamol (n=1) | P (n=1) |
| Other | 2 (4) | Enalapril (changed to other brand because of suspected side effects) (n=1) | P/G (n=1) |
| | | Metformin (compliance discussed) (n=1) | P (n=1) |

*≥1 Actions could be recorded for a single patient.
F, planned follow-up; G, treatment goal; N/A, not applicable; P, drug treatment problem.

Interestingly, the proportion of patients with suboptimal drug treatment, irrespective of priority, was similar for patients with and without a procedure code explicitly stating that a medication review had been performed over the last year. However, patients with such a procedure code had a greater burden of disease, also illustrated by the higher number of drugs in the medication list.[32] Thus, the treatment in multimorbid patients can be expected to be more complex, and it is reassuring that drug treatment quality is acceptable to the same extent also in these patients. Although this pilot study does not allow conclusions, one may speculate that performed medication reviews may have contributed to appropriate drug treatment in multimorbid patients requiring complex treatment considerations.

Patients as defined in the Swedish regulations to receive a medication review at least annually, that is, patients ≥75 years of age with ≥5 drugs in the medication list, had this procedure code documented to a large extent. The adherence to this regulation, despite the absence of an established association between age and drug treatment quality,[18] and the limitations of cut-offs of number of drugs in the medication list to reflect quality of drug treatment,[33] may illustrate that monetary compensations can direct healthcare efforts to some extent. However, the evidence for effects of pay for performance policies in improving quality of care and health outcomes is limited.[34] Indeed, a recent study, investigating the impact of pay for performance regarding the procedure code 'medication review' found no apparent association between this code and the prevalence of PIMs, although the number of patients with such a code increased.[35]

The actions performed within the documented medication reviews were quite few, as only 1 out of 14 drugs were acted on. This finding may give the impression that medication reviews by the attending physician is a minor effort. However, as a medication review implies a medical assessment, ascertaining that the drug treatment is reasonable given the current health status, the decision not to change a drug may be just as important. Moreover, the basic part of a medication review, the medication reconciliation, was not captured in the present study.

We were happy to find that the problem underlying the treatment action was documented in the medical records in all but one case. Indeed, such information is important in transitions of care. Nevertheless, our results also suggest that documentation is an area with potential for improvement; the goal of the drug treatment change was only stated in about half of the cases, and the planned follow-up in a minority.

In conclusion, the results of this pilot study suggest that current prescribing practices in older people in primary care is in general appropriate, especially when medical priorities are considered. Efforts for rational use of medicines over the last decades may have contributed to this finding, including easily available information on recommended drugs,[36 37] computerised decision support,[38 39] and indicators of prescribing quality for benchmarking

and educational purposes.[21 40] Indeed, although the latter may reflect quality of drug treatment to a limited extent, they may be valuable in particular for junior physicians, as they reflect expert opinions on current best practice for an average older patient. We also want to emphasise that, although our findings indicate that drug treatment in older people in primary care may be quite good, endeavours for improvements are always desirable. Our results, reflecting drug treatment quality from a broad and clinically relevant perspective taking individual factors and medical priorities in the complex clinical situation into account, may be useful in sample size considerations for future non-randomised and randomised studies, focusing to factors associated with prescribing practices and interventions.

**Contributors** NPL contributed to study design, acquisition of data, analysis and interpretation of results. SMW contributed to study design, analysis and interpretation of results. Both authors drafted and revised the article, and approved the final manuscript.

**Funding** This work was supported by the Swedish Research Council; the Swedish state under the agreement between the Swedish government and the county councils, the ALF-agreement; and the Department of Research and Development in Södra Älvsborg.

**Competing interests** None declared.

**Patient consent for publication** Not required.

**Ethics approval** The study was approved by the Regional Ethical Review Board in Gothenburg, Sweden (DRN: 1046-15). Informed consent was waived in the approval.

**Provenance and peer review** Not commissioned; externally peer reviewed.

**Data sharing statement** The datasets generated and analysed during the current study are not publicly available due to Swedish data protection laws. The data can be shared with authorized persons after approved application from the Regional Ethical Review Board in Gothenburg (https://www.epn.se/en/start/the-organisation/).

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
