## [Reviewer comments · BMJ Open]

ARTICLE DETAILS

TITLE (PROVISIONAL)	Quality of prescribing in older people from a broad family physician perspective: A descriptive pilot study
AUTHORS	Parodi Lopez, Naldy; Wallerstedt, Susanna

VERSION 1 – REVIEW

REVIEWER	May Nawal Lutfiyya University of Minnesota, Minneapolis, MN USA
REVIEW RETURNED	13-Nov-2018

GENERAL COMMENTS	This is an interesting paper BUT I am not convinced by the paper as it is currently written that a physician perspective is absent from earlier similar studies. Also this is a pilot study conducted in a single clinic over the course of a single month. How does such a study speak to a broader audience?
--

REVIEWER	Cheryl Sadowski University of Alberta, Canada
REVIEW RETURNED	20-Nov-2018

GENERAL COMMENTS	Summary: This study used a sample of a primary care practice in Sweden to describe the prescribing practices by physicians in this practice, and to suggest that this information can be used as baseline for future studies on medication appropriateness. Overall this is an interesting study and the challenge of addressing potentially inappropriate medication use is particularly a challenge for primary care physicians. There are a few areas that require further review and consideration. Terminology – please use “older adults” or “geriatric patients” instead of ‘elderly’, and please use the same term consistently. Please review for flow of language as it appears some sections were written by different authors. Title – describing this as a pilot study is appropriate. I would suggest a term such as ‘quality’ or ‘appropriate prescribing’ or ‘geriatrics’ or ‘older adults’ might be more descriptive and lead people to this article who are interested in this issue. The ‘physician perspective’ implies this might be physician feedback or a qualitative methodology, but it appears to be simply describing physician activity related to medication reviews in seniors. Abstract Although it is noted later in the paper, it would be important to note that the findings relate only to the physicians, as an interprofessional team was not involved in care.
---

	Some medications listed may be available only in Sweden or the EU (e.g. propiomazine). It is fine to list this, but internationally it might lose some meaning if that drug is not used universally. Conclusion – the statement that drug treatment is generally appropriate is true if the standard is to have the majority of patients on appropriate medications, but 51% is a low bar. Conclusion – the final statement about using the study for future sample size calculations appears to be confusing because there is nothing earlier in the abstract that the study is for that purpose. That should be put into the purpose of the study, in the first sentence of the abstract. Article Summary The first bullet should specify if the priorities taken into account are the prescribers', patients' priorities, or both. Background Second paragraph is helpful in order to understand the context in Sweden. It may be helpful to define the terms for the health professionals in the clinic. For example, a medical practitioner who is fully licensed vs a GP? As a reviewer from North America I'm not quite sure the credentials of pre-reg house officer (a student trainee?). It's important because it appears not all the participants were licensed physicians, so their views on medication prescribing and appropriateness may not be very mature or reflect practicing physicians. For Page 5 row 44 – please include references supporting that low prescribing quality is a problem. Methods Setting – explain the rationale for the selection of this one particular primary care practice vs others in Sweden. Timeline – a study done over just 1 month of data collection is extremely short. Were all 10 physician participants there for that entire month? Are there usually more physicians? Why was this one month chosen, versus a sampling over a longer period of time? Were the medication counts and changes measured during this month alone? Explain if the physicians knew their work was being monitored. Identification of PIMs – it appears the tools such as STOPP, EU7-PIM, etc. were used. The measures seem appropriate for the context and are valid. Please specify if the physician was allowed to identify other medications as inappropriate based on judgment versus following a criteria. Identification of PIMs – the article describes numerous tools, but the abstract is a bit misleading by listing just the categories – could possibly include the number of criteria used. Assessments – it appears each physician participant made decisions about each medication for changes, and there was no second review to confirm this assessment. Assessments – page 9 line 37-40 – “For indicators based on kidney function, we estimated the glomerular filtration rate (eGFR) according to the Cockcroft-Gault equation.” Note that this equation estimates CrCl not eGFR. Defining a medication review – which standard was used for this review, or in Sweden are physicians allowed to conduct it any way they chose, and still bill the code? Was a process such as medication reconciliation done, or a structured medication review?
--	--

	Medication list – were medications listed by the patient, from a national pharmacy record, or other method? How were doses gathered? Application of screening tools – this is well explained, but there does need to be clarification if the process for assessments was done with a computer program, built into the electronic health record, or if this was done from memory, hard copy? Chart/EMR documentation – how were the changes by the physician recorded? Was the entire chart open for review for this study, or was one particular section targeted for this information? Results The different levels of physician trainees are described in methods, but the analysis is done as a group. Given the small number of physicians this is not powered to look at differences, but I wonder if this is problematic in that there was quite a range of experience amongst this group. If the findings were not different by prescriber, then maybe this is just a limitation. In Table 1 please explain if those with impaired cognition are a unique billing code, or why you chose to separate that group of patients. Also you list those with a procedure code in the past year in separate columns, but do not compare the differences. Table 1 – explain multi-dose drug dispensing – does this mean a duration of medications provided to the patient, a measure of polypharmacy? Table 1 – morbidity – was this the reason for referral or assessment visit? Why are these listed – simply the most common in the sample? Table 1 – under Number of Drugs – is this the average number of oral medications/patient at the time of that one visit, pre or post visit? The statistic of >5 drugs should be a separate denominator as it appears to be 54 patients and 30 patients in the 2 groups? Page 12 – line 37 – what does ‘partly overlapping’ mean, and how was this statistic determined? Discussion The opening sentence that 6/10 patients had appropriate care is technically true, but a rather low bar – 40% of seniors are still receiving ‘risky’ medications? The assumption appears to be that physicians did a good job in this study of identifying problems and fixing them, but we are not certain the quality of their work, or if there were problems not identified that they should have caught. Interpretation – page 16, line 29-30 – is this a good thing that physicians may provide lower figures? Please explain how the setting of Sweden provides an ideal context for this study to provide a baseline rate to be used by larger scale studies. Limitations – a major limitation appears to be the lack of patient input into the medication review. For patient-centred care the appropriateness criteria need to take individual patient perspectives into account. In other settings pharmacists and nurses provide medication reviews, therefore it seems limited that these perspectives were not considered either. Limitations – explain how a study of 10 physicians, some with varying degrees of training, can be generalized to practice in Sweden, the EU, or more broadly internationally. Limitations – note that non-oral medications were excluded. A number of these could be inappropriate, and lead to medication
--	---

	burden and cost, polypharmacy, errors, etc, which is inappropriate. This also highlights that the medication review was not a medication review as defined in other standards, where it must include all the patient's medication. The second paragraph talks about dosage adjustment, and it assumes this was a good change, but further explanation is required because we are not provided doses in the Results, and the doses may be patient dependent (e.g. renal clearance).
--	--

VERSION 1 – AUTHOR RESPONSE

Authors response to reviewer 1:

1. This is an interesting paper BUT I am not convinced by the paper as it is currently written that a physician perspective is absent from earlier similar studies. Also this is a pilot study conducted in a single clinic over the course of a single month. How does such a study speak to a broader audience?

Authors' response: We are grateful that the reviewer pointed out the need to clarify the novelty of this pilot study. Compared with the extensive previous literature on quality of prescribing, we take a broader approach and priorities in the complex clinical situation into account. This has been clarified in the revised manuscript (abstract, aim; page 6, lines 85-86, 99-100; page 16, lines 300-301). Further, the title has been revised. Thereby this pilot study reflects a realistic picture on the quality of drug treatment in health care, providing clinically relevant data for power calculations in full-scale studies on prescribing quality. This has been emphasized in the revised discussion (page 20, lines 399-400)

Authors response to reviewer 2:

1. Terminology – please use “older adults” or “geriatric patients” instead of ‘elderly’, and please use the same term consistently. Please review for flow of language as it appears some sections were written by different authors.

Authors' response: We appreciate this suggestion from the reviewer. ‘Elderly’ has been replaced by “older people” throughout the revised manuscript. Further, we have carefully reviewed the text for flow of language.

2. Title – describing this as a pilot study is appropriate. I would suggest a term such as ‘quality’ or ‘appropriate prescribing’ or ‘geriatrics’ or ‘older adults’ might be more descriptive and lead people to this article who are interested in this issue. The ‘physician perspective’ implies this might be physician feedback or a qualitative methodology, but it appears to be simply describing physician activity related to medication reviews in seniors.

Authors' response: We agree with the reviewer that “quality” and “older” may increase the visibility of the article for those with interests within the field. In order to clarify the content of the article, we have changed the title to “Quality of prescribing in older people from a broad family physician perspective: A descriptive pilot study”.

3. Abstract. Although it is noted later in the paper, it would be important to note that the findings relate only to the physicians, as an interprofessional team was not involved in care.

Authors' response: In the revised manuscript, we have clarified that a broad family physician perspective is the scope of the paper, this profession holding the full licence to prescribe (title; abstract; page 5, line 69; page 6, lines 99-100). Therefore, we do not focus on the team behind. Nonetheless, we agree that team contributions are valuable, and mention this as a limitation in the revised manuscript (page 16, line 316). For instance, nurses may contribute important relevant information for medication reviews regarding the medications used and patient symptoms.

4. Some medications listed may be available only in Sweden or the EU (e.g. propiomazine). It is fine to list this, but internationally it might lose some meaning if that drug is not used universally.

Authors' response: As the reviewer points out, the availability of drugs may vary between countries. In the revised manuscript, we describe the drugs more thoroughly to facilitate for an international readership (page 2, line 34; page 12, lines 236-238).

5. Conclusion – the statement that drug treatment is generally appropriate is true if the standard is to have the majority of patients on appropriate medications, but 51% is a low bar.

Authors' response: The overall physician assessment showed that 45 (37%) had suboptimal treatment, 13 (29%) of whom with a suggested change in drug treatment of high priority. This implies that 78 (63%) patients had an appropriate treatment, and even more so if low priority changes are not counted. We have revised the conclusion and the results text (page 3, line 42; page 12, 232).

6. Conclusion – the final statement about using the study for future sample size calculations appears to be confusing because there is nothing earlier in the abstract that the study is for that purpose. That should be put into the purpose of the study, in the first sentence of the abstract.

Authors' response: Thank you for this suggestion. In the revised abstract, this aim of the pilot study is included in the objectives.

7. Article Summary. The first bullet should specify if the priorities taken into account are the prescribers', patients' priorities, or both.

Authors' response: We appreciate this observation from the reviewer and have clarified in the first bullet that medical priorities are in focus (page 4, lines 5). This has also been clarified in the abstract (page 2, lines 25-26; page 3, line 41), aim (page 6, line 100), the methods (page 9, lines 163-165), and the conclusion (page 20, lines 397-398).

Background

8. Background. Second paragraph is helpful in order to understand the context in Sweden. It may be helpful to define the terms for the health professionals in the clinic. For example, a medical practitioner who is fully licensed vs a GP? As a reviewer from North America I'm not quite sure the credentials of pre-reg house officer (a student trainee?). It's important because it appears not all the participants were licensed physicians, so their views on medication prescribing and appropriateness may not be very mature or reflect practicing physicians.

Authors' response: In the revised manuscript, we have clarified the physician categories to facilitate the interpretation for the international readership (page 6, line 108; page 7, lines 109-110).

9. For Page 5 row 44 – please include references supporting that low prescribing quality is a problem.

Authors' response: We appreciate that the reviewer observed this omission. In the revised text, we cite relevant references (page 5, line 78).

Methods

10. Methods. Setting – explain the rationale for the selection of this one particular primary care practice vs others in Sweden.

Authors' response: In this pilot study, we chose a centre that could be representative for primary health care centres. Serving a population of 10,000 individuals, including two nursing homes, and staffed by 10 physicians, an acceptable variation in prescribing could be expected. Nevertheless, this centre may differ from others regarding a number of aspects, for example population covered and prescribing traditions. This has been further emphasized as a limitation of the study (page 16, lines 311-313).

11. Methods. Timeline – a study done over just 1 month of data collection is extremely short. Were all 10 physician participants there for that entire month? Are there usually more physicians? Why was this one month chosen, versus a sampling over a longer period of time? Were the medication counts and changes measured during this month alone?

Authors' response: In the sample size considerations for this pilot study, we considered that including all patients fulfilling our inclusion criteria during one month, anticipated to amount to about a hundred, would yield useful results for the power calculations in the full-scale study, i.e. reasonably certain prevalence figures on suboptimal drug treatment as well as information on the prevalence of documented medication reviews (page 10, lines 196-201). The data were collected retrospectively. At the start of data collection, we chose the most recent month for which the notes in the medical records were expected to be completed. The study did not focus on the performance of individual prescribers. Therefore, capturing data at the physician level was not approved by the ethical review board, and information on the contribution to the results of each physician cannot be calculated. However, all physicians at the centre were on duty the month in question.

Regarding the medication lists, the counts refer to medications for regular use or as needed, marked as current medications at the end of the physician consultation. This has been clarified in the revised manuscript (page 8, lines 134-137). The changes in the medication list included those that were done during the physician consultation as described in the methods section (page 8, lines 145-146).

12. Methods. Explain if the physicians knew their work was being monitored.

Authors' response: The physicians in the health care centre were not informed beforehand that their work would be monitored. This has been clarified in the revised manuscript (page 7, lines 123-124).

13. Identification of PIMs – it appears the tools such as STOPP, EU7-PIM, etc. were used. The measures seem appropriate for the context and are valid. Please specify if the physician was allowed to identify other medications as inappropriate based on judgment versus following a criteria.

Authors' response: The family physician performing the assessments in this study was allowed to identify inappropriate drug treatment not covered by the screening tools. This has been clarified in the revised manuscript (page 10, lines 190-191).

14. Identification of PIMs – the article describes numerous tools, but the abstract is a bit misleading by listing just the categories – could possibly include the number of criteria used.

Authors' response: We appreciate this observation from the reviewer and have added this information in the abstract (page 2, line 26).

15. Assessments – it appears each physician participant made decisions about each medication for changes, and there was no second review to confirm this assessment.

Authors' response: In this study, we retrospectively reviewed the prescribing practices during one month in a primary care centre. This has been clarified in the revised abstract (line 20), methods (page 7, line 122). One assessor made the assessments, and this is mentioned as a limitation of the study (page 16, lines 315-316).

16. Assessments – page 9 line 37-40 – “For indicators based on kidney function, we estimated the glomerular filtration rate (eGFR) according to the Cockcroft-Gault equation.” Note that this equation estimates CrCl not eGFR.

Authors' response: We have revised the text regarding renal function (page 10, lines 186-189).

17. Defining a medication review – which standard was used for this review, or in Sweden are physicians allowed to conduct it any way they chose, and still bill the code? Was a process such as medication reconciliation done, or a structured medication review?

Authors' response: In the second paragraph of the background, we have clarified that medication reviews are regulated by national authorities (page 5, lines 69-72). As described in this paragraph, medication review implies that the physician reconciles the drug treatment and assesses the benefit/risk for all drugs separately and combined, to ascertain that the treatment is reasonable given the current health status.

18. Medication list – were medications listed by the patient, from a national pharmacy record, or other method? How were doses gathered?

Authors' response: All data were obtained from the electronic medical records, including electronic prescriptions with detailed information on the prescribed product, the dose, if intended for regular use or as needed, the duration of the treatment, the reason for the treatment, and the amount prescribed. This has been clarified in the revised manuscript (page 8, lines 134-136).

19. Application of screening tools – this is well explained, but there does need to be clarification if the process for assessments was done with a computer program, built into the electronic health record, or if this was done from memory, hard copy?

Authors' response: The assessments for PIMs/PPOs were performed manually using a hard copy of the criteria. This has been clarified in the revised text (page 9, line 177; page 10, lines 189-190).

20. Chart/EMR documentation – how were the changes by the physician recorded? Was the entire chart open for review for this study, or was one particular section targeted for this information?

Authors' response: The assessor had access to the entire medical records including all electronic prescriptions. Using these sources, changes in drug treatment could be detected. This has been further described in the revised manuscript (page 8, lines 147-148).

Results

21. The different levels of physician trainees are described in methods, but the analysis is done as a group. Given the small number of physicians this is not powered to look at differences, but I wonder if this is problematic in that there was quite a range of experience amongst this group. If the findings were not different by prescriber, then maybe this is just a limitation.

Authors' response: We agree that physicians' experience may be of importance for the quality of prescribing. As we wanted to evaluate drug treatment in primary care in general, a variation in experience was desirable. Indeed, investigating the impact of experience on prescribing was not the scope of the study, and the prescribing physicians were not the research subjects. Consequently, such data capture and analyses were not approved by the ethical review board. Nevertheless, investigating these aspects could be of interest in future studies.

22. In Table 1 please explain if those with impaired cognition are a unique billing code, or why you chose to separate that group of patients. Also you list those with a procedure code in the past year in separate columns, but do not compare the differences.

Authors' response: Those with impaired cognition do not yield a unique billing code. We obtained information on characteristics which we considered could be of importance for quality of prescribing and performance of medication reviews. In this pilot study, we do not perform any statistical comparisons as the aim was to provide descriptive data for sample size considerations in future studies. The study was not powered for comparisons.

23. Table 1 – explain multi-dose drug dispensing – does this mean a duration of medications provided to the patient, a measure of polypharmacy?

Authors' response: We appreciate this comment from the reviewer and have clarified that multi-dose drug dispensing implies that the patient receives machine-dispensed unit bags with drugs that should be ingested concomitantly, the system being intended for patients who have difficulties in handling their medications (page 7, lines 126-127).

24. Table 1 – morbidity – was this the reason for referral or assessment visit? Why are these listed – simply the most common in the sample?

Authors' response: The listed morbidities are those which were most common in the sample. This has been clarified in the table.

25. Table 1 – under Number of Drugs – is this the average number of oral medications/patient at the time of that one visit, pre or post visit? The statistic of >5 drugs should be a separate denominator as it appears to be 54 patients and 30 patients in the 2 groups?

Authors' response: In Table 1, values are presented as median (interquartile range) or number of individuals (percentage) as described below the table. Among those with a procedure code recorded regarding a medication review being performed (n=65), 54 (83%) persons had five or more drugs in the medication list. Among those without such a procedure code (n=58), 30 (52%) persons had five or more drugs in the medication list. The number of drugs refer to the post-visit counts. This has been clarified in the revised table.

26. Page 12 – line 37 – what does 'partly overlapping' mean, and how was this statistic determined?

Authors' response: Similar criteria can be found in several screening tools, for example potential overtreatment with proton pump inhibitors and long-acting benzodiazepines. This has been clarified in the revised manuscript (page 10, lines 185-186). As the criteria are often not exactly phrased the

same way but may concern the same drug/drug group, they may partly overlap. In order not to introduce arbitrary assessments, we chose not to combine the criteria, but to apply each set separately and simply count each criteria once.

Discussion

27. The opening sentence that 6/10 patients had appropriate care is technically true, but a rather low bar – 40% of seniors are still receiving ‘risky’ medications?

The assumption appears to be that physicians did a good job in this study of identifying problems and fixing them, but we are not certain the quality of their work, or if there were problems not identified that they should have caught.

Authors’ response: We agree with the reviewer that 40% of seniors still receive risky medications. However, given that the risky medications may not always be highest priority during the family physician visit and the fact that they were mainly sedatives/anxiolytics and proton pump inhibitors, drugs which are not easily withdrawn, we believe that the physicians do a quite good job in the complex clinical situation.

28. Interpretation – page 16, line 29-30 – is this a good thing that physicians may provide lower figures?

Authors’ response: We realize that this sentence was unclear and have revised it (page 17, lines 335-337). In the sentence, we discuss that PIMs/PPOs may not be relevant when taking the overall clinical picture into account, that is, from a broad family physician perspective.

29. Please explain how the setting of Sweden provides an ideal context for this study to provide a baseline rate to be used by larger scale studies.

Authors’ response: We do not claim that this setting provides an ideal context for any full-scale study. However, we consider the figures presented more clinically relevant than those frequently reported previously within this field of research. Indeed, our assessments of drug treatment quality are relevant from a medical perspective as a broad family physician perspective is applied, taking individual factors and priorities in the complex clinical situation into account. The figures may therefore provide a realistic basis for power calculations in full-scale studies. This has been clarified at the end of the discussion (page 20, lines 399-400).

30. Limitations – a major limitation appears to be the lack of patient input into the medication review. For patient-centred care the appropriateness criteria need to take individual patient perspectives into account. In other settings pharmacists and nurses provide medication reviews, therefore it seems limited that these perspectives were not considered either.

Authors’ response: We agree with the reviewer that patient input is important in medication reviews. As this was a descriptive, retrospective, medical records-based study, we do not know to what extent the patients were actively involved in the medication reviews. The same goes for the potential involvement of other professionals. According to Swedish regulations, as described in the introduction, performing medication reviews is the responsibility of the physician. The physician may be informed on medications and symptoms from, for example, nurses in nursing homes. However, only physicians have a full licence to prescribe. This has been clarified in the revised manuscript (page 5, line 69).

31. Limitations – explain how a study of 10 physicians, some with varying degrees of training, can be generalized to practice in Sweden, the EU, or more broadly internationally.

Authors' response: The number of physicians is only mentioned to help the reader to understand the setting. In the manuscript, we discuss the limitations regarding external validity (page 16, lines 312-313). Nevertheless, as mentioned above, this study contributes realistic and clinically relevant figures on quality of drug treatment, valuable for sample size considerations in full-scale studies.

32. Limitations – note that non-oral medications were excluded. A number of these could be inappropriate, and lead to medication burden and cost, polypharmacy, errors, etc, which is inappropriate. This also highlights that the medication review was not a medication review as defined in other standards, where it must include all the patient's medication.

Authors' response: Only medications without systemic effects were excluded e.g. carbomer eye drops. All other medications, oral and non-oral, were included. This is described in the methods section (page 8, line 138).

33. The second paragraph talks about dosage adjustment, and it assumes this was a good change, but further explanation is required because we are not provided doses in the Results, and the doses may be patient dependent (e.g. renal clearance).

Authors' response: A dosage adjustment implies that the dose is adjusted according to the medical assessment of the specific patient. For example, the metformin dose was increased due to high glycated hemoglobin. In this sample, no drug changes were made due to reduced kidney function. This has been clarified in the revised manuscript (page 14, line 272-273; page 16, lines 296).

VERSION 2 – REVIEW

REVIEWER	Cheryl Sadowski University of Alberta, Canada
REVIEW RETURNED	24-Apr-2019

GENERAL COMMENTS	This draft is very well written and is acceptable in its current form. I have a few additional considerations: the inclusion of 'broad family physician perspective' through the title and the entire document is a bit overstating, as we don't really know what perspective the physicians used, what they were thinking, process, etc. We hope it was broad because they were directed to be broad. Summary bullet point - the tools used were numerous but were they really 'extensive'? These tools are mostly lists, and are helpful, but not always in depth. Article summary - the third bullet point doesn't quite seem to fit, as the limitations are in the Discussion. I'd rather see a bullet summarizing the findings, such as the percent of patients who had appropriate or inappropriate drugs, or which intervention was most common. Methods - suggest declaring if any tools were used for the life expectancy estimate, or if that was physician estimate based on experience.
--

	Results - page 14 lines 260-264 need some follow-up in Discussion, because it appears from these statements that the medication reviews don't have much impact. Conclusions - the expectations that >50% of people with appropriate regimens is acceptable seems a bit of a low bar, but the article generally seems quite positive about the 60% or so that had safer regimens. We aren't actually sure what the goal should be, or what is reasonable. For example, vaccinations should be over 95% in the population, or chronic disease management (e.g. hypertension) should be about 60-80% for reaching a target. Without a target accepted in the literature this study is important, but I'm not sure I agree that it is impressive that almost half of patients still have some high risk meds.
--	--

VERSION 2 – AUTHOR RESPONSE

Authors response to reviewer 2:

1. The inclusion of 'broad family physician perspective' through the title and the entire document is a bit overstating, as we don't really know what perspective the physicians used, what they were thinking, process, etc. We hope it was broad because they were directed to be broad.

Authors' response: As discussed in the previous review round, primarily in response to comments by reviewer 1, the broad and medically relevant perspective constitutes the main contribution of this manuscript, compared with previous scientific literature within the field. Therefore, the perspective was broad because it was directed to be broad. We agree that "broad" is appearing too often in the manuscript, and have omitted the word where appropriate not to overstate this aspect (page 2, line 31; page 6, line 84; page 9, line 159; page 17, line 335).

2. Summary bullet point - the tools used were numerous but were they really 'extensive'? These tools are mostly lists, and are helpful, but not always in depth.

Authors' response: We agree with the reviewer and have replaced "extensive" with "numerous" (page 4, line 54; page 16, line 303).

3. Article summary - the third bullet point doesn't quite seem to fit, as the limitations are in the Discussion. I'd rather see a bullet summarizing the findings, such as the percent of patients who had appropriate or inappropriate drugs, or which intervention was most common.

Authors' response: According to the editorial request, we have changed the title for this section to "Strengths and limitations of this study". In this context, we consider the third point appropriate and hope that the reviewer agrees with us (page 4, lines 55-57).

4. Methods - suggest declaring if any tools were used for the life expectancy estimate, or if that was physician estimate based on experience.

Authors' response: Thank you for this suggestion. In the revised methods, we have clarified that this estimate was based on the assessor's experience (page 9, line 165).

5. Results - page 14 lines 260-264 need some follow-up in Discussion, because it appears from these statements that the medication reviews don't have much impact.

Authors' response: We agree that the results highlighted by the reviewer may be worth further attention, i.e. that the proportion of patients with suboptimal drug treatment, irrespective of priority, was similar for patients with and without a procedure code explicitly stating that a medication review had been performed over the last year. Therefore, we have added this aspect in the discussion (page 18, lines 351-352). The lack of impact of medication reviews is discussed on page 18 (lines 351-359 and 368) and page 19 (lines 369-371).

6. Conclusions - the expectations that >50% of people with appropriate regimens is acceptable seems a bit of a low bar, but the article generally seems quite positive about the 60% or so that had safer regimens. We aren't actually sure what the goal should be, or what is reasonable. For example, vaccinations should be over 95% in the population, or chronic disease management (e.g. hypertension) should be about 60-80% for reaching a target. Without a target accepted in the literature this study is important, but I'm not sure I agree that it is impressive that almost half of patients still have some high risk meds.

Authors' response: We agree with the reviewer that there is still room for improvements, as emphasized in the text (page 20, lines 394-396). However, given that the majority of patients with suboptimal treatment, according to the family physician assessment, had low priority problems and only a few patients had medications where an immediate handling was suggested, we believe that a positive approach is justified. Nevertheless, we have tempered the conclusion paragraph in the revised manuscript (page 19, lines 387-389).